# Advancement in long-distance bird migration through individual plasticity in departure

Jesse R. Conklin [1✉], Simeon Lisovski [2✉] & Phil F. Battley [3✉]

Globally, bird migration is occurring earlier in the year, consistent with climate-related changes in breeding resources. Although often attributed to phenotypic plasticity, there is no clear demonstration of long-term population advancement in avian migration through individual plasticity. Using direct observations of bar-tailed godwits (*Limosa lapponica*) departing New Zealand on a 16,000-km journey to Alaska, we show that migration advanced by six days during 2008–2020, and that within-individual advancement was sufficient to explain this population-level change. However, in individuals tracked for the entire migration (50 total tracks of 36 individuals), earlier departure did not lead to earlier arrival or breeding in Alaska, due to prolonged stopovers in Asia. Moreover, changes in breeding-site phenology varied across Alaska, but were not reflected in within-population differences in advancement of migratory departure. We demonstrate that plastic responses can drive population-level changes in timing of long-distance migration, but also that behavioral and environmental constraints en route may yet limit adaptive responses to global change.

---

[1] Conservation Ecology Group, Groningen Institute for Evolutionary Life Sciences (GELIFES), University of Groningen, Groningen, The Netherlands. [2] Alfred Wegener Institute, Helmholtz Centre for Polar and Marine Research, Telegrafenberg, Potsdam, Germany. [3] Wildlife and Ecology Group, School of Agriculture and Environment, Massey University, Palmerston North, New Zealand. ✉email: conklin.jesse@gmail.com; simeon.lisovski@awi.de; P.Battley@massey.ac.nz

The current pace and scale of environmental change tests the capacity of organisms to adapt. For example, climate change is altering the timing of annual cycles in animals and plants worldwide[1,2]. In general, the timing of bird migration and breeding are advancing[3,4], as expected with earlier availability of seasonal resources for reproduction[5]. However, weak or inadequate phenological responses are commonly observed[6], as are general declines in migratory populations[7,8], implying limitations to the capacity for adaptation. For long-distance migrants, this is complicated by changes that may vary geographically in degree or direction[9], or include interactions with non-climate-related effects such as direct habitat alteration by humans[10]. Currently, the specific mechanisms driving adaptive population changes are poorly understood.

Observed population-level changes in avian migration phenology are generally attributed to phenotypic plasticity[11], but the additional or interacting role of evolutionary responses are rarely tested and difficult to exclude[12,13]. Individuals are often relatively consistent in migration timing[14], but both observational and experimental studies show that individual timing is flexible and responsive to environmental conditions, body state, and social context[15–17]. Observed magnitudes of plasticity may be sufficient to account for documented population-level phenology changes[18], particularly in long-lived species that may use lifelong experience to adjust annual routines[19,20]. However, data to directly address this question are scarce, due to the difficulty of simultaneously collecting long-term population-wide information and repeated individual data for any stage of migration. One such study found that arrival at breeding grounds did not change within adult individuals over time, but that population advancement occurred through the recruitment of young individuals with increasingly early migration timing[21]. To our knowledge, there has been no demonstration of long-term, directional population change in migration timing resulting from individual plasticity.

The annual migration of bar-tailed godwits (*Limosa lapponica baueri*) includes three flights of 6000–12,000 km each, including two of the longest non-stop flights recorded in birds[22,23]. Across the Alaska breeding range (Fig. 1), there is a latitudinal cline in which northern-breeding godwits are smaller and migrate later than southern breeders[24,25]; the timing differences reflect the increasingly later snowmelt, and thus the availability of tundra breeding sites, at higher latitudes. At non-breeding sites in New Zealand, annually consistent individual differences in departure on the 10,000-km non-stop flight to the Yellow Sea region of Asia are maintained across a one-month period[26,27], and are generally retained throughout the northward migration[24,27]. In the non-breeding season, bar-tailed godwits present a unique opportunity to directly observe both population-level and individual behavior, due to their relatively large size, high site fidelity and longevity, gregarious use of open habitats, and conspicuous migration departure during daylight hours[15].

Here, we use 13 years of directly observed migratory departures by bar-tailed godwits from a small, intensively-monitored non-breeding site in New Zealand (Fig. 1) to: (1) describe the magnitude of population-level change in the initiation of migration, and (2) assess the relative contributions of individual plasticity and between-individual differences to the population trend. By tracking a subset of individuals with light-level geolocators for the complete northward migration, we quantify the extent to which between-year changes in departure persist to later stages, including the timing of arrival and breeding in Alaska. To evaluate expected responses, we describe phenology changes in the species' Alaska breeding range during the same period, using remotely sensed environmental data. We demonstrate that directional, population-level change in timing of bird migration can occur predominantly through within-individual changes:

across 13 years, bar-tailed godwits advanced their departure from New Zealand by nearly 0.5 days per year, and within-individual advancement alone was sufficient to explain the trend.

## Results

During 2008–2020, we documented all migratory departures from the Manawatu River estuary in New Zealand, which included 10–14 flocks containing 128–251 total godwits per year. Over this period, mean northward departure advanced at −0.484 d/yr ± SE 0.035 ($R^2 = 0.073$, $F_{1,2421} = 192.7$, $p < 0.0001$), from ca. 21 to 15 March (Fig. 2a).

Based on individually-identifiable godwits in these flocks (124 marked individuals observed in 3–13 years each), plastic responses were sufficient to explain the population-level advancement in departure: the overall within-individual trend (−0.428 d/yr ± SE 0.100) was not distinguishable from the between-individual trend (−0.463 d/yr ± SE 0.082, $p = 0.78$; Fig. 2b and Supplementary Table 1a). Within-individual trends ranged from −6.5 to +5.5 d/yr, and were advancing in 69% of individuals (86 of 124; Supplementary Fig. 1). Despite the overall advancement, consistent between-individual differences were retained (individual repeatability of departure during 2008–2020: $r = 0.782 ± SE 0.021$, $F = 23.0$, d.f. = 123, $p < 0.0001$).

Population-level advancement could potentially occur through a changing composition of the study population; for example, through lower survival or lower recruitment of later-migrating (i.e., smaller, northern-breeding) birds across the study period. However, we found no evidence of a changing proportion of northern- and southern-breeders across annual samples: mean body size was constant over time (Supplementary Fig. 2) and the likelihood of returning the next year was unrelated to departure date (generalized linear model: slope = −0.0005, $Z = −0.28$, d.f. = 746, $p = 0.78$).

During the same period (2008–2020), advancement in the timing of snowmelt (date when 33% snow-free) within the entire Alaska breeding range (−0.651 d/yr ± SE 0.001; Supplementary Table 2) was similar in magnitude to the advancement in godwit departure from New Zealand, although characterized by large between-year variation. However, changes varied regionally: snowmelt advanced at −1.926 d/yr ± SE 0.003 in southern (<64°N) Alaska, compared to −0.235 d/yr ± SE 0.002 in northern (>64°N) Alaska (Fig. 3c and Supplementary Table 2). Advancement of spring green-up (date of greatest increase in Normalized Difference Vegetation Index, NDVI) was also much greater in southern (−2.296 d/yr ± SE 0.031) than in northern Alaska (−0.467 d/yr ± SE 0.014; Fig. 3d and Supplementary Table 2).

To determine whether advancement in departure from New Zealand differed by breeding destination, we reduced the data set of directly observed departures to 34 individuals with known breeding locations, determined from geolocator-tracking in four years of the study (2008, 2009, 2013, 2014; Fig. 3a and Supplementary Fig. 3). For this sample, within-individual advancement of departure from New Zealand during 2008–2020 was similar for godwits that bred in northern Alaska (−0.500 d/yr ± SE 0.206, $n = 15$) and southern Alaska (−0.522 d/yr ± SE 0.209, $n = 19$; Fig. 3b and Supplementary Table 1b).

Geolocator-tracking (50 complete northward migration tracks of 36 individuals) also showed that the advancement in departure from New Zealand did not lead to earlier arrival in Alaska (Fig. 4 and Supplementary Table 3). During 2008–2014, despite advancement of the first flight from New Zealand to the Yellow Sea (departure: −0.845 d/yr ± SE 0.85; arrival: −0.885 d/yr ± SE 0.267; both $p ≤ 0.005$), the subsequent flight to Alaska did not significantly advance (departure: −0.333 d/yr ± SE 0.257; arrival: −0.205 d/yr ± SE 0.233; both $p ≥ 0.20$). This difference is

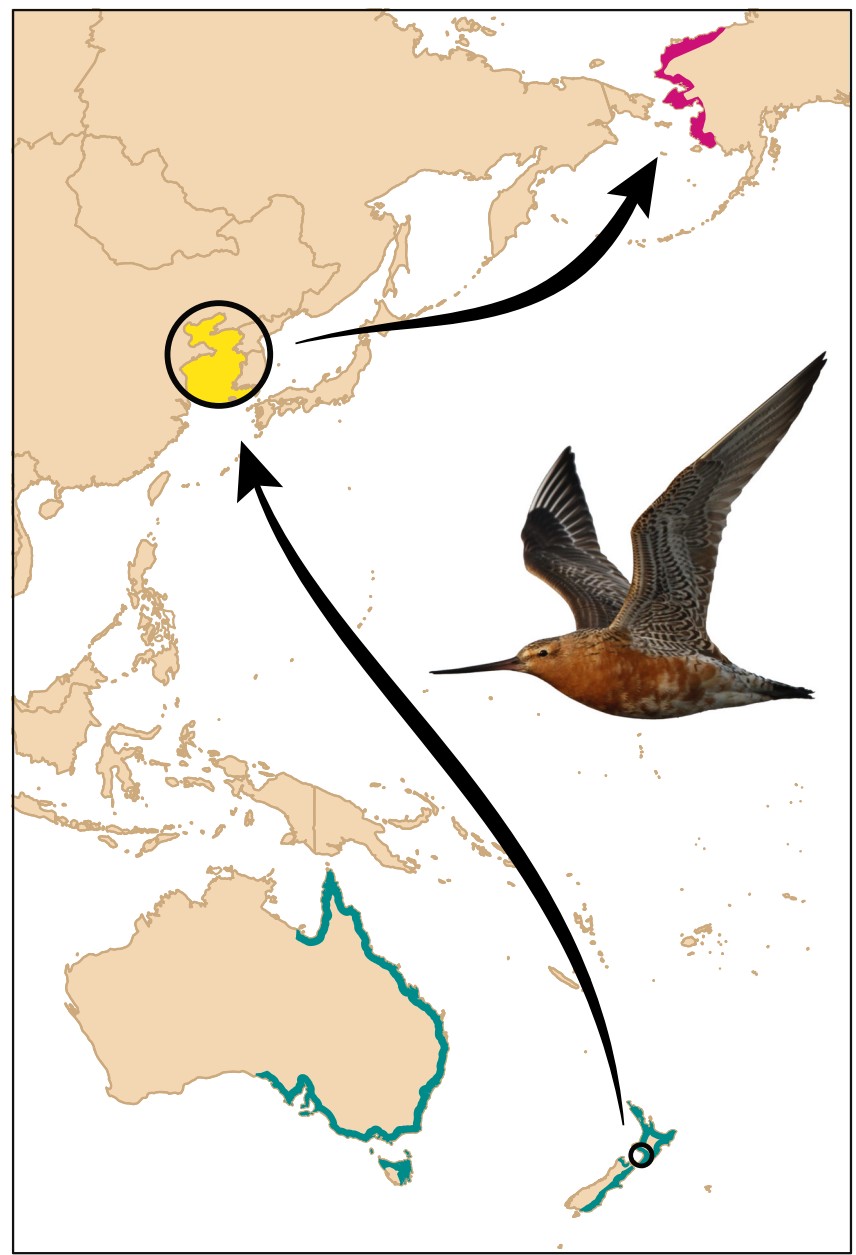

**Fig. 1 Northward migration of bar-tailed godwits from the non-breeding range in New Zealand and Australia (in blue) to breeding areas in Alaska (in pink) in two non-stop flights, with a stopover in the Yellow Sea (in yellow).** Field site in New Zealand indicated by a circle. Map made with Natural Earth. Photo: P. Battley.

explained by the increasing duration of the stopover in the Yellow Sea (+0.693 d/yr ± SE 0.302, $p = 0.026$). The date that godwits started incubating their first clutch ($n = 41$ nests of 29 individuals) did not change across 2008–2014 (+0.066 d/yr ± SE 0.402, $p = 0.87$). Differences in regional trends in the timing of snowmelt and NDVI in Alaska during the shorter period of geolocator-tracking (2008–2014) were similar to the longer-term (2008–2020) trends (Fig. 3c, d and Supplementary Table 2).

## Discussion
Our findings contrast with the expectation that inter-generational or evolutionary processes are required to explain observed phenological changes in bird migration[28,29]. In our analysis of migratory departure from New Zealand, we aimed to explicitly separate phenotypic flexibility (i.e., reversible plastic changes

within a post-development adult[30]) from between-individual shifts in the population over time[31], which may occur through developmental plasticity or micro-evolution through selection on different phenotypes. The statistically indistinguishable within- and between-individual slopes (Fig. 2b and Supplementary Table 1) indicate that no selection or generation shift is required to explain the advancement in migration timing of bar-tailed godwits.

It is commonly claimed that weak or insufficient responses by migratory birds to advancing phenologies at breeding grounds (in terms of timing of arrival or breeding) may reflect an inherent lack of flexibility in the initiation of migration[32,33]. This claim is particularly invoked for longer-distance migrants, which may face greater temporal or physiological constraints and stronger canalization of migratory behavior[11]. The magnitude of the advance in the departure of godwits from New Zealand is comparable to or

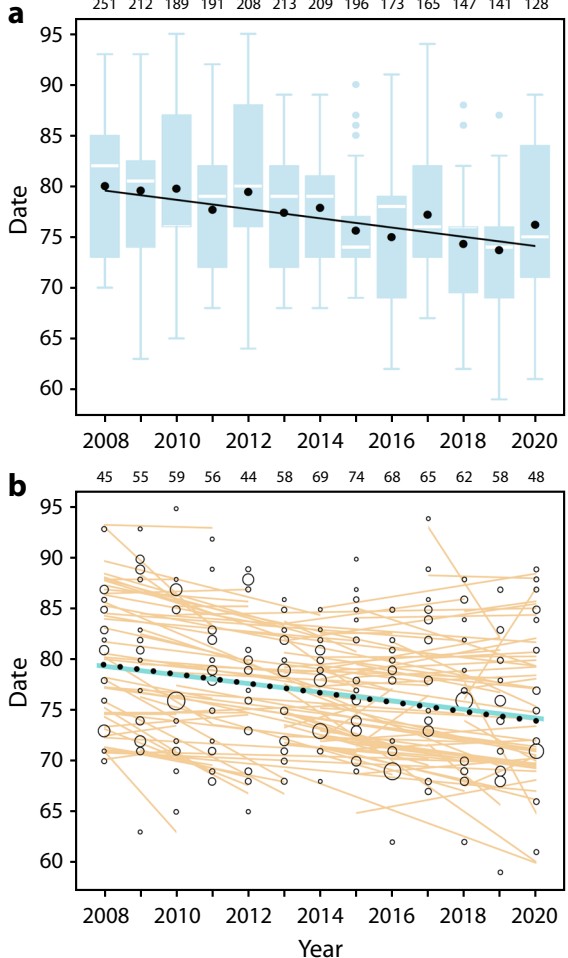

**Fig. 2 Advancement in godwit migratory departure from New Zealand during 2008–2020, at the population and individual levels. a** Population-level advancement in departure. In blue: distribution of departures (*n* = 128–251 godwits per year, total = 2,423 departures; sample sizes above panel); boxplots indicate 25–75th percentile range with median departure date (white lines); whiskers extend to 1.5× interquartile range; outliers (dots) beyond this. In black: mean departure dates (filled circles) plus linear slope (solid line). Date: 1 = 1 January. **b** Within-individual advancement of departure date (*n* = 124 marked individuals observed in 3-13 years each; sample sizes above panel). Open circles = all departures; size scaled to the proportion of total migrating individuals that year (0.006–0.422 per day). Orange lines indicate within-individual slopes. Overall advancement was shown by the solid blue line (within-individual slope) and black dotted line (between-individual slope); see Supplementary Table 1a.

greater than recent population-level changes observed at the passage or terminal sites of both short- and long-distance migrants (advances of typically 0.1–0.5 d/yr[34–36]). Therefore, our results do not support limited plasticity in the initiation of migration as an explanation for insufficient population responses to advancing phenology of breeding resources. This may be an unexpected finding in an extreme long-distance migrant that shows unusually high individual repeatability in migration timing[14,26,27], but demonstrates that repeatability does not strictly imply consistency[27], and that neither precludes potential flexibility in new circumstances.

Although developmental or evolutionary changes are unnecessary to explain our results, we cannot exclude that some also occurred during our study. A longer time series is required to specifically address generational change in this species, as our

study period did not involve a complete turnover of individuals. On average, godwits in this population will migrate to Alaska nine times (adult annual survival ca. 88%[37]), and some individuals surpass 25 years of age. In our study, most individuals (93%) were of unknown age, having been first marked as adults (≥3 years old), and 19 individuals contributed 10–13 years of data. Therefore, we do not know whether the degree of plasticity we observed would persist and be sufficient to address multi-decadal change in this population[4], or would potentially reach a functional limit in a longer-term study[38].

Advancement of migration timing is generally assumed to reflect a response to global climate change, particularly at breeding grounds, and it is possible that godwits are departing New Zealand earlier in an attempt to track advancing conditions in Alaska. Indeed, snowmelt across their breeding range as a whole advanced at a similar rate (−0.65 d/yr) as the departure from New Zealand (−0.48 d/yr). However, earlier departure by godwits did not lead to earlier arrival in Alaska (Fig. 4), and the more rapid advancement of breeding conditions in southern Alaska was not reflected in a relatively greater response by southern-breeding godwits (Fig. 3b). This suggests that the advancement in migration timing was not simply, and perhaps not at all, a response to conditions in Alaska.

After departing New Zealand, migrating godwits stage on the shores of the Yellow Sea in East Asia, a region threatened by habitat loss and decreased food supplies. Intertidal mudflats in the Yellow Sea, used by this and many other migratory shorebird populations on the East Asian-Australasian Flyway, have been significantly lost and degraded in recent decades[39], coincident with general population declines[40] and decreasing survival, including in bar-tailed godwits[37,41]. At the most important northward stopover site for Alaska-breeding godwits, the Yalu Jiang National Nature Reserve in northeast China, benthic prey and godwit intake rates plummeted during 2011–2013 and prey levels have not recovered subsequently[42]. In our geolocator-tracked godwits, earlier departure from New Zealand in 2013–2014 versus 2008–2009 was completely countered by a prolonged stopover in Asia. This raises two alternative, non-mutually-exclusive explanations for the observed change in New Zealand.

The first is that advancing departure from New Zealand is a response to advancing breeding phenology in Alaska, but that deteriorating conditions in Asia prevented a similar advancement of the second flight to Alaska. Lower habitat quality at staging sites could decrease fueling rates[43,44] and thus increase the required length of stay in Asia. In this scenario, the consistent timing of arrival in Alaska is maladaptive, resulting from reversible-state effects[45] of disruptive conditions along the migratory route. A second alternative is that advancing departure from New Zealand is actually a response to conditions in Asia rather than Alaska, as a mechanism to allow more time to refuel in Asia while maintaining a consistent arrival date in Alaska. In this scenario, the increased length of stay in Asia is the adaptive response, successfully preventing later arrival in Alaska.

With current data, we cannot exclude either of these scenarios, nor can we rule out an additional role of changing conditions in New Zealand. However, we find this last possibility unlikely to drive a directional trend in timing, because previous work has shown departure from New Zealand to be generally insensitive to seasonal carry-over effects[46], and days with advantageous wind conditions for migration are quite common[15]. Currently, we have no evidence that the length of stay in New Zealand, which is approximately six months, has changed over our study period. Further, any resource-based hypothesis for advancing departure would need to reconcile earlier fueling for a 10,000-km non-stop flight with both a decreasing study population (Fig. 2a) and declining benthic prey resources at the study site (PFB & JRC unpubl. data).

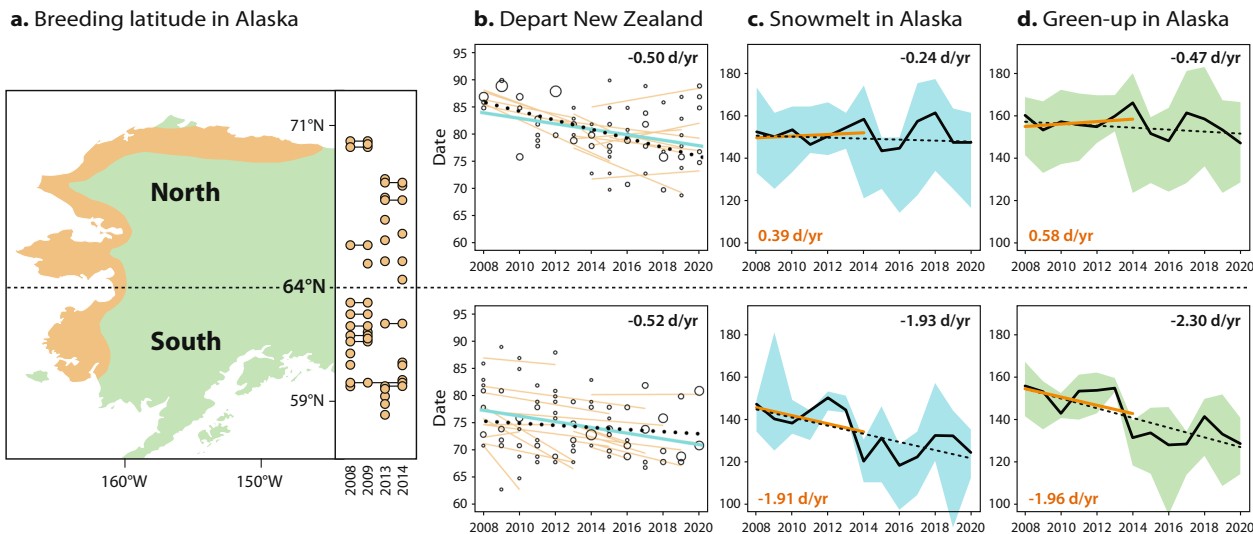

**Fig. 3 Changes in the initiation of godwit migration and breeding phenology according to destination in Alaska. a** Breeding latitude of 36 geolocator-tracked individuals (orange dots; lines indicate individuals tracked in multiple years) in four years of tracking (2008, 2009, 2013, 2014; n = 12–13 per year), and assigned to Alaska region (North >64°N, n = 16; South <64°N, n = 20). The known breeding range is shown in orange. Map made with Natural Earth. **b** Within- and between-individual advancement of directly observed departure date from New Zealand during 2008–2020 for individuals with known breeding locations in northern (upper; n = 15 individuals observed in 3–13 years each) and southern Alaska (lower; n = 19). Date: 1 = 1 January. Figure details as described in Fig. 2b. The within-individual slope is indicated in the panel (see Supplementary Table 1b). **c** Timing of snowmelt (date when 33% snow-free) in the Alaska breeding range north (upper) and south (lower) of 64°N. **d** Start of spring green-up (date of greatest increase in Normalized Difference Vegetation Index, NDVI) in the Alaska breeding range north (upper) and south (lower) of 64°N. For **c** and **d**, black lines indicate median date across region, and shaded areas indicate 95% quantile range. Dotted lines indicate the 13-year trend (2008–2020; slope indicated in the panel), and orange lines indicate the trend during the period of geolocator-tracking (2008–2014).

If individual godwits are advancing their migration timing in New Zealand based on changing conditions in Asia or Alaska, it implies the existence of feedback mechanisms that allow modification of circannual cycles according to past events or conditions (i.e., learning). For godwits, there are no conceivable within-year cues on the non-breeding grounds in New Zealand for stopover or breeding conditions in the northern hemisphere (ca. 10,000 and 16,000 km along the migration route, respectively). However, as the average godwit will experience numerous migrations, adults have ample opportunity to use previously experienced conditions at staging and breeding sites to inform migratory decisions[19,47], enabling intermediate-term (i.e., year-to-year) plastic responses. Because godwits migrate in flocks and are highly social in the non-breeding season, individuals may be additionally influenced by the behavior and experiences of others[20,48]. Also, young individuals departing New Zealand on their first northward migration typically join flocks of experienced adults; this may represent a mechanism for incremental, inter-generational change, without the need for replacement through natural selection[20]. Such flexible modifications to endogenous annual rhythms may be common in long-lived species, but less important for short-lived species experiencing only 1–3 migrations. Precisely what mechanisms allow long-distance migrants to track long-term environmental changes, which are often more stochastic than incremental (e.g., Fig. 3c, d), remain poorly understood[49].

Currently, we lack data to address whether the relatively constant timing of Alaska arrival has affected fitness through decreased reproductive success. However, we can derive some insights from geolocator-derived nest incubation behavior (Fig. 4). After arrival in southwest Alaska, bar-tailed godwits spend up to two weeks refueling at coastal sites before moving to tundra breeding sites[22,24], and this period represents an opportunity to gain information about local environmental conditions and potentially compensate for sub-optimal arrival timing.

According to geolocator data, in 2008, 2009, and 2013, northern-breeding godwits started incubation ca. 18 days (annual means 16.5–20.5 days) after arrival in Alaska, compared to 23 days (annual means 20.3–27.5 days) for southern-breeding birds. In these years, the timing of snowmelt in both regions was close to the 13-year trend (Fig. 3c). However, in 2014, snowmelt was unusually late in northern Alaska, and unusually early in the south (Fig. 3c). As expected, northern breeders delayed the start of incubation (mean 24.0 days after arrival), while southern breeders started incubation much sooner after arrival (mean 15.7 days). This implies flexibility in the system to respond to local phenology after arrival in Alaska when necessary, but a longer time series, including a greater range of environmental fluctuations, is required to understand the limits of this flexibility.

The surprisingly large reaction norm in New Zealand departure may imply that other stages of the migration would be similarly plastic, given the right environmental conditions. However, it is also conceivable, particularly in trans-hemispheric migrations in which distant locations are linked by one or few flights[23], that different stages of the migration have been shaped by very different regimes of selection, and thus respond differently to change[50]. Such modularity is demonstrated by different endocrine and time-keeping mechanisms among annual-cycle stages such as molt, migratory fueling, and breeding[51,52], and in the common observation that temporal variation decreases with increasing proximity to breeding areas[53,54]. It is also clear that en route conditions can de-couple timing of departure and arrival, such that variation observed at the migratory destination may not reflect the degree of potential flexibility in earlier stages[55,56], as we found in this study.

The expectation that long-distance migrants may be limited in their capacity to respond to change stems, in part, from the understanding that migration timing is set by endogenous programs that are entrained by daylength[32,57], in birds without relevant within-season cues about destination conditions[49].

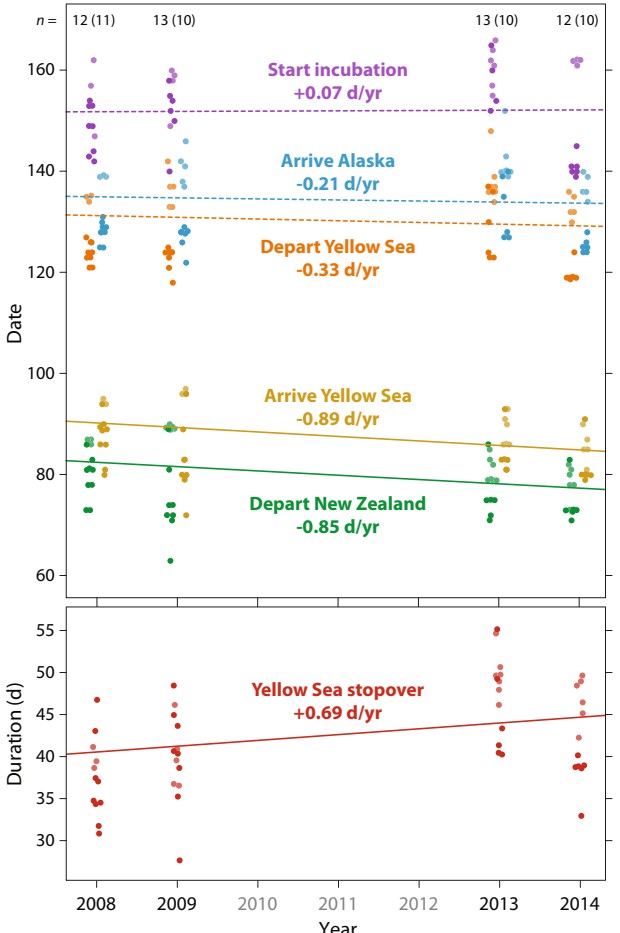

**Fig. 4 Northward migration and breeding phenology for geolocator-tracked godwits in 2008, 2009, 2013, and 2014.** Points for overlapping stages and individuals are offset for clarity. Within each stage (distinguished by color), dark and pale dots indicate individuals breeding in southern (<64°N) and northern (>64°N) Alaska, respectively (Fig. 2a). Lines indicate significant (solid) and non-significant (dashed) slopes during 2008–2014, derived from linear mixed models (see Supplementary Table 3). Numbers across the top of the plot indicate the number of tracked birds per year (smaller sample of incubating birds in parenthesis). Date: 1 = January.

While there are clearly fundamental endogenous components to migration, there is increasing evidence for substantial capacity for change within certain timing windows that are presumably set by heritable processes and photoperiod[58]. In New Zealand, there is latitudinal variation in the timing of godwit migration, with birds in the southern part of the non-breeding range (with longer photoperiods) departing significantly earlier than those further north[59]. Across all studied sites, larger (southern-breeding) birds migrated earlier on average than smaller (northern-breeding) birds, implying both external (photoperiodic) control of the population migration timing, and internal (i.e., endogenous program) control of individual schedules. The current study indicates that, overlaid on this framework of individual- and population-specific timing, adult godwits can advance their migration timing, presumably in response to past conditions throughout the annual cycle.

In summary, we have demonstrated population-level advancement in the initiation of avian migration that is explicable by within-individual changes in post-development adults. This shows that, at decadal scales and for long-lived species, evolutionary or

other inter-generational mechanisms are not necessarily required for population adaptation to phenological changes in annual conditions and resources. However, a lack of earlier arrival at breeding grounds suggests that flexibility at departure can still be disrupted by conditions or control mechanisms that vary across stages of migration[9,51,55]. Although most studies of phenological change in migratory birds are focused on adaptation to climate effects on the breeding grounds, our results additionally highlight the potential for phenological responses to human-induced changes en route, which may be largely independent of climate effects[60]. It is not clear that the observed degree of phenotypic flexibility, or subsequent evolution of phenological traits[61], will be sufficient to track current and future environmental change[6,62,63]. However, it may offer some optimism regarding timely adaptation by migratory and other apparently time-constrained species, provided the critical habitats and resources they require are preserved[56].

## Methods

**Study site and population.** The Manawatu River estuary (40.47°S, 175.22°E; Fig. 1) is a small (ca. 1 × 2 km) intertidal mudflat area on the west coast of the North Island, New Zealand. Since 2006, we have captured bar-tailed godwits by cannon-net or mist-net, and marked individuals with a numbered metal band, plus either a unique combination of the white flag and four colorbands, or an engraved white flag with a field-readable three-digit alphabetical code. During 2008–2020, 35% (range: 23–48% per year) of adult (i.e., migratory) individuals were marked in the estuary's highly site-faithful population of 170–280 godwits.

At capture, godwits were aged (≥/<3 years) by plumage or state of primary feather molt; because godwits ≥3 years cannot be precisely aged, the great majority of birds in our studies are of unknown age. Due to sexual dimorphism (females > males), most (89%) individuals were sexed by bill length (length of exposed culmen: >99 mm = female; <88 mm = male), but intermediate birds (88–99 mm) cannot be sexed by this measure[25]; however, conspicuous dimorphism in plumage before departure from New Zealand allowed unambiguous sexing of the remaining individuals. In this population, size and migration timing in both sexes vary along a latitudinal cline in the Alaska breeding range (60–71°N; higher latitude = smaller and later migration)[24,25].

**Directly observed migratory departures.** The local godwit population is highly approachable and can usually be observed entirely from one of several vantage points along the perimeter of the mudflat. Each year, a single observer (occasionally two) monitored the flock daily from ca. 1 March until all migratory individuals had departed (29 Mar–5 Apr). During surveys, the observer watched and listened for migratory behavior, which included distinct vocalizations and low circling flights expressing intent to depart. Flocks typically engaged in conspicuous calling, preening, bathing, and short exploratory flights for 0.5–4 h before actual departure[15]; during this time, the observer recorded all marked individuals involved, using a 20–60× spotting scope and digital camera with a 400 mm lens. All flying flocks were watched and/or photographed until they resettled or disappeared from sight. After each departure, the observer quickly surveyed the estuary to account for all remaining marked godwits. In addition, we conducted daily high-tide surveys to confirm the size and composition of the remaining flock; the average daily resighting probability of marked godwits was 88% (range 51–100% per individual) before departure.

We recorded departures of 44–81 marked individuals per year (174 total individuals). In 86% of these departures, the bird was directly observed preparing and/or departing with an observed flock, and therefore exact time and flock size were known. The remaining individuals were assigned a departure date based on the last day they were recorded at the estuary. For 10% of cases, this coincided with an observed departing flock of partially or completely unknown individual composition; therefore, we considered a time and flock size to be known. For the remaining 4% of cases, the individual's disappearance coincided with a decrease in local flock size unexplained by observed departures; we considered these departures unobserved, and calculated flock size based on successive high-tide counts. 98% of observed departures occurred during 13:00–21:00, and unobserved departures likely occurred at night or early morning. By including unobserved departures, annual totals represent a virtually complete accounting of flocks and individuals migrating from the site.

We are confident that observed movements out of the estuary represented a migratory departure from New Zealand. Site fidelity of marked godwits is extremely high for the entire non-breeding season (September–March), and nonmigratory movements in and out of the estuary were rare. Departures were easily distinguished from local movements by both altitude and direction; departing flocks always flew NW/NNW and slowly ascended toward the ocean before disappearing from view. Furthermore, for all departures captured by both direct observation and geolocator tracking (n = 51; see below), conductivity (wetness) data recorded by geolocators confirmed ca. 7-day dry period (indicating

a non-stop flight to Asia) starting on the same day (0–2.5 h difference) as the observed departure[64].

All statistics were calculated in the software R 3.5.1[65]. Using departures of all godwits ($n$ = 128–251 total individuals in 12–17 flocks on 10–14 days per year), we analyzed phenology change across years using a linear model in the R package lme4 v.1.1.21[66].

Including only individuals observed departing New Zealand in ≥3 years ($n$ = 124 individuals; 44–74 per year), we distinguished the contributions of within- and between-individual variation using within-subject centering[31] in a linear mixed model in lme4. To explore within-population variation in these effects, we performed the same analysis separately for the two regions in the Alaska breeding range (North, South; Fig. 3), including 34 individuals with known breeding locations from geolocator tracking (see below).

To quantify the proportion of individuals (≥3 observations) that showed an advancement in their departure timing, we obtained individual posterior distributions via direct simulations of 1000 values from the joint posterior distribution of ordinary linear model parameters using the function sim from the R package arm v.1.11-2[67]. The 2.5% and 97.5% quantiles of the simulated values across individuals were used to define the range of slopes. The proportion of negative slopes within-individual simulations was used to estimate the percentage of individuals showing advanced departure dates.

We calculated repeatability (intra-class correlation coefficient, $r$ ± SE)[68] of departure date from New Zealand for 124 individuals observed 3–13 years each.

**Geolocator tracking and analysis.** In 2008–2009 and 2013–2014, a subset of individuals was tracked for the full migration to Alaska using light-level geolocators. The units (1.0–1.5 g) were mounted on a colorband on the tibia; attachment plus all individual markings represented 1–2% of lean body mass. We deployed geolocators in March 2008 ($n$ = 17 MK14, British Antarctic Survey, UK), October 2008 ($n$ = 19 MK14), February 2013 ($n$ = 15 MK4093, Biotrack, UK, and $n$ = 24 Intigeo-C65K, Migrate Technology, UK), and November 2013 ($n$ = 22 Intigeo-C65K) and recaptured birds during the following non-breeding season(s) to retrieve the units. This resulted in the following number of complete northward tracks (including breeding location; see below): 12 in 2008, 13 in 2009, 13 in 2013, and 12 in 2014; 25 individuals were tracked in 1 year, 11 in 2 years, and 1 in 3 years.

Godwits depart New Zealand near the austral autumnal equinox, when light-level geolocation is least effective for describing movements, particularly for flights in a north-south direction[69]. However, geolocators also recorded conductivity (i.e., contact with salt water); wet-dry transitions can thus identify extended periods of flight between intertidal habitats with high precision[64]. From conductivity data, we derived four parameters for each northward track: dates of departure from New Zealand (NZdep), arrival and departure from the Yellow Sea region of Asia (YSarr, YSdep), and first arrival in Alaska (AKarr). Most godwits spend up to two weeks in coastal SW Alaska before moving to breeding sites[22,24], but some individuals show no discernible shift in light or conductivity data during this period. Therefore, we did not analyze arrival at the ultimate breeding site. We derived a fifth timing parameter, duration of the stopover in the Yellow Sea (YSdur), as the difference (in days) between YSarr and YSdep.

Because the geolocators were leg-mounted, light-level data in Alaska also indicated periods of nest incubation, when the unit was shaded by the sitting bird[24,27]. During the breeding season, geolocators registered nights as regular, clearly demarcated periods of darkness 0–4.5 h in length; these did not appear at all if birds were north of 64°N. Days appeared as continuous light, irregularly broken by a brief (<1 h) shading events, most likely corresponding to behaviors such as wading or sitting. Within 6–25 days of apparent arrival on breeding grounds, most godwits (41 of 50 total tracks) displayed a conspicuous pattern of incubation, in which semi-regular shading events of 4–13 h were overlaid on the day/night pattern for periods up to 25 d. Bar-tailed godwits incubate tundra nests bi-parentally, and so both sexes demonstrate this pattern. We considered the first day of this period to be the start of incubation (IncSt), and analyzed this timing parameter along with the five migration parameters (above).

For godwits tracked in 2008–2009 using MK14 geolocators, breeding locations are published previously[24] (Supplementary Fig. 3a). We used BASTrak[70] software (British Antarctic Survey, UK) to produce twice-daily location estimates, and calculated mean latitude and longitude (plus 95% range of estimates) during stationary periods in the breeding season, after removing light-dark transitions clearly affected by behavioral shading events such as incubation. Based on periods when the bird's true location was known (at the deployment site in New Zealand, and in some cases additionally in Asia), we used sun angles of 3.0–3.6° below the horizon to represent twilight for calibration of light-level data for location estimates. For three individuals that traveled north of the Arctic Circle, narrow-range light sensitivity of MK14 geolocators precluded location estimation in 24-h daylight; however, longitude estimates immediately before and after the breeding season were east of 160°W, strongly indicating breeding destinations on the central North Slope of Alaska. Therefore, we assumed a breeding latitude of 70.2°N (the mid-point of the known breeding range in this region; see Supplementary Fig. 3a) for these individuals. For individuals tracked in multiple years ($n$ = 6), the location derived from the year with the best available data (i.e., longer uninterrupted stationary breeding period) is shown in Supplementary Fig. 3a.

For godwits tracked in 2013–2014 using Intigeo-C65K geolocators, which recorded full-range light levels, we estimated breeding locations using the R package *PolarGeolocation* v.0.1.0[71]. To calibrate the maximum light curve and the error distribution of light recordings from the maximum light curve, we used the longest possible period for each bird, i.e., all days when the bird was at the Manawatu River estuary (range 38–161 days; mean = 127 days). Next, we selected a time period during the breeding season that included a clear pattern of breeding (incubation inferred from shading patterns); although including incubation increases uncertainty in the breeding-site estimate, we chose such periods to ensure that we did not include periods spent away from the breeding site in the location estimation. The duration of the period used to estimate breeding locations was 12–48 days (mean = 28 days). *PolarGeolocation* estimates a gridded likelihood surface from which we can derive confidence levels of our best location estimate; these confidence intervals depend on the extent of the grid. To make the confidence estimates comparable across individuals, we ran a first estimate using a grid with 50 km resolution and a radius of 1500 km around the approximate center of the breeding range (162°W, 62°N). Next, we re-centered the mask around the best estimate and re-ran the simulation using the same radius of 1500 km, and the same likelihood contour level of 0.05 to describe the confidence intervals for all birds (Supplementary Fig. 3b). For individuals tracked in multiple years ($n$ = 4), the location derived from the first year of tracking is shown in Supplementary Fig. 3b.

To estimate breeding sites from four Biotrack MK4093 geolocators in 2013, we performed the simple threshold approach. First, we defined sunrise and sunset times using a light intensity threshold of 2 arbitrary units, and used the periods the individual bird was at the deployment site in New Zealand for calibration and calculation of a reference zenith angle[69] that was then used to estimate locations via the *thresholdPath* function in the R Package *SGAT* v.0.1.3[72]. All loggers showed dark periods during the breeding season (i.e., remained south of the Arctic Circle), allowing estimation of locations throughout the year. We extracted the locations for the period including incubation and calculated the median location and the 95% credibility interval.

Bar-tailed godwits breeding in northern and southern Alaska differ in size and migration timing[24,25], and potentially experience very different temporal trends in conditions during the breeding season. Therefore, tracked individuals were grouped into two regional groups, comprising those breeding >64°N (North) or <64°N (South). Individuals tracked in multiple years and/or with different geolocator types allow assessment of repeatability of breeding location estimates; however, we note that the uncertainty measures we show in Supplementary Fig. 3 for the three estimation methods are not strictly comparable. For six individuals with breeding locations derived in both 2008 and 2009 from MK14 geolocators, location estimates differed by 24–124 km, respectively. For four individuals tracked with Intigeo units in both 2013 and 2014, location estimates differed by 50–190 km, respectively. For one individual tracked with both MK14 and Intigeo loggers, the best estimates for 2008 and 2014 differed by approximately 78 km. These differences fall within the expected error of geolocation, and in no case affected an individual's assignment to the northern or southern groups. The final set of tracked godwits with known breeding latitude included 36 individuals (North = 16, South = 20; Fig. 2a and Supplementary Fig. 3).

To describe the rate of change across 2008–2014 in six migration and breeding phenology parameters (NZdep, YSarr, YSdep, AKarr, IncSt, and YSdur; see above), we used linear mixed models including individual as a random effect and breeding region (North or South) as a fixed factor (Fig. 4, Supplementary Table 3). To account for unequal samples of northern and southern breeders across years, we initially included the interaction of year*breeding region in the models; this interaction factor was non-significant for all phenology parameters ($p$ = 0.08–0.85), and so we excluded it from the final models.

**Environmental data and analysis.** In Arctic-breeding shorebirds, the timing of breeding closely follows the retreat of snow-cover from tundra nest sites in the spring[73]. We compared two indices of breeding phenology (timing of snowmelt and spring green-up), summarized separately for the two regions (North, South) of the Alaska breeding range of bar-tailed godwits. We did this for two temporal periods: one encompassing the entire study period (2008–2020), and a shorter-term related directly to the period in which we tracked individuals with geolocators (2008–2014).

*Snowmelt.* Remotely sensed IMS Daily Northern Hemisphere Snow and Ice Analysis data for the period 2008–2020 on a scale of 4 × 4 km were downloaded[74]. Next, grid cells falling in the subspecies' breeding range were extracted. Then, data from cells that did not include measures of sea ice were modeled using a maximum likelihood fit (*mle2* function from R package *bbmle* v.1.0.23.1) of the asymmetric Gaussian model function[75] with a binomial error distribution. The model fit is a compromise between the first snow-free day and the day when the pixel remained snow-free in late spring and summer (for more details and comparison between different snow-melt indices see Supplementary Methods). Using these year-specific fits, the date of snowmelt was then determined as the date on which the fitted curve predicted 1/3 of the cell area to be snow-free.

*Normalized Difference Vegetation Index (NDVI).* From the NOAA STAR Vegetation Health Product data set, weekly noise-removed NDVI data on a scale of 4 ×

4 km grid cells were downloaded for the period 2008–2020[76]. Next, cells located in the subspecies' breeding range were selected for further analysis. To identify the annual day when the vegetation starts to grow (green-up day), a penalized cubic smoothing spline was fitted to the yearly NDVI values, allowing daily interpolation and the calculation of when the curve exceeds 15% of the greatest amplitude during spring[77]. To ensure a robust fit of the smoothing spline and to remove potential artifacts of single pixels, the NDVI values of the surrounding pixels within a radius of 15 km were taken into account. However, the distance to the focal cell was used to weigh the importance of the model fit (with a Gaussian decline over distance with a standard deviation of 4 km). Because the presence of snow and ice can significantly affect NDVI values, the model was used to interpolate daily NDVI values only for days that were snow-free in at least one of the pixels within the 15 km radius. Next, the day of the maximum NDVI was established, and the day when the increasing NDVI curve (start to maximum) exceeded 15% of the amplitude was extracted as the NDVI start date. More detailed information and a comparison between the estimated NDVI start date and the higher spatial resolution MODIS derived green-up day (also based on a 15% threshold) are in Supplementary Methods.

Snowmelt and NDVI start were highly correlated across the breeding range ($t_{160221} = 664.95$, $p < 0.001$, $R^2 = 0.856$) as well as in the northern ($t_{120852} = 494.19$, $p < 0.001$, $R^2 = 0.817$) and southern ($t_{39367} = 285.65$, $p < 0.001$, $R^2 = 0.821$) regions separately.

**Ethics statement.** Fieldwork was conducted with Massey University Animal Ethics Committee approval (#07/163, 12/90, and 16/117) and appropriate New Zealand Department of Conservation permits (Banding permit 2007/39, 35503-FAU, 38111-FAU).

**Reporting summary.** Further information on research design is available in the Nature Research Reporting Summary linked to this article.

## Data availability

The behavioral data generated in this study are available on Zenodo (https://doi.org/10.5281/zenodo.5016733). The environmental data used in this study are available from NSIDC (https://doi.org/10.7265/N52R3PMC) and NOAA (ftp://ftp.star.nesdis.noaa.gov/pub/corp/scsb/wguo/).

## Code availability

The code for environmental analyses performed in this study is available on Zenodo (https://doi.org/10.5281/zenodo.5025715).

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

## Acknowledgements

We thank colleagues for helpful discussions and comments on previous drafts: Christiaan Both, Bart Kempenaers, Jelle Loonstra, Theunis Piersma, Eldar Rakhimberdiev, Joost Tinbergen, Mo Verhoeven, and especially Yvonne Verkuil. We thank David Melville, Adrian Riegen, Rob Schuckard, and many volunteers for trapping and banding assistance, and Iris Bontekoe for fieldwork in 2018. The collection of long-term departure data was supported by Chris & Neville Hopkins, David & Lucile Packard Foundation, Dobberke Foundation for Comparative Psychology, Manawatu Estuary Trust, Marsden Fund (Royal Society of New Zealand), Massey University Doctoral Scholarship, New Zealand Department of Conservation, Ornithological Society of New Zealand, Pacific Shorebird Migration Project, Pūkorokoro Miranda Naturalist's Trust, and Royal Netherlands Academy of Arts & Sciences (KNAW). We acknowledge support by the Open Access Publication Funds of Alfred-Wegener-Institut Helmholtz-Zentrum für Polar- und Meeresforschung and of Massey University.

## Author contributions

J.R.C. and P.F.B. conceived the study. J.R.C. collected and analyzed the observational data. J.R.C. and P.F.B. deployed and retrieved geolocators. All authors analyzed geolocator data. S.L. analyzed the environmental data. J.R.C. wrote the manuscript with contributions from S.L. and P.F.B. All authors reviewed and edited the manuscript.

## Competing interests

The authors declare no competing interests.
