## [Peer Review File · Nature Communications]

Reviewer comments, first round –

Reviewer #1 (Remarks to the Author):

This study is one of the first to robustly show that individual birds are changing their phenology of migration and departing from wintering areas earlier in recent years, coinciding with the patterns expected by climate warming and the need to arrive at breeding sites earlier. Bar-tailed godwits are long lived species and differences in phenology could result from individual or population responses to environmental change (previous studies have shown disparate results). This is a long term study quantifying both population and individual phenology changes using colour rings observations and geolocator information. The authors show that departure dates from wintering sites in New Zealand advanced 0.46d/y at the population level and this was consistent with the individual responses (0.42d/y). This study shows that in long lived species individuals have the capacity to adjust their migration phenology throughout their lives, this is an interesting result with implications for conservation as adaptation to environmental change can occur at the individual level rather than only through natural selection at the population level. A few questions remain (see below).

The population and individual data on departure from NZ was obtained from a long term dataset (13 years of data) and the results are robust. These results clearly show that phenology changes are occurring at the population and individual level, although there is high variability within individual trends “Within-individual trends ranged from –6.5 to +5.5 d/yr, and were advancing in 69% of individuals”, suggesting that there could be differences between northern and southern Alaskan breeders. The authors attempted to explain this variability based on geolocator information that provide full migration phenology information for individual godwits.

The geolocator information complements the individual analysis obtained for the departure from NZ dates and provides further insights into the timing of different stages of the migration. This dataset was obtained over 4 years (2 early and 2 later years) spread apart, hence is limited, and stochastic environmental variability could influence the results obtained. Furthermore, the proportion of south/north Alaskan breeding birds changes from the first to the second period of tracking which could influence the results. Lack of advancement in arrival date and start of breeding could be a reflection of a higher proportion of Northern breeders in the second period of tracking. To test if this is the case, the authors could have included the interaction between year and breeding location in the models presented in table S3 (this may be stretching the model due to the small sample size, if this is the case a bootstrapping approach could be used).

In the discussion the authors give to hypotheses for the increasing stopover duration observed at the Yellow sea site. These are plausible but it would be good to know more about the length of the stay in New Zealand, is this also shifting? This may help understand the changes in phenology and raise alternative hypothesis for the early departure from NZ and longer stopover.

Another hypothesis: Godwits are social birds, if southern breeders are advancing their breeding date and migrating early, this could induce northern breeders to abandon wintering areas early (staying behind in smaller groups could lead to increased predation risk), but lead them to stay longer at the stopover locations compensating for the early departure. Stopover locations with larger numbers of birds from different wintering and breeding locations may provide good conditions to wait for the “right departure time”. If only southern breeders are advancing their migratory phenology, this

would not be noticeable after the departure from NZ (again this could be tested by including the interaction between year and breeding location in the models presented in table S3). Overall, I enjoyed reading this manuscript, it is clearly structured and provides new insights on how changes in migratory phenology operate in long lived species, at the population and individual level. The results presented are robust but the geolocator information could have been examined for interactions between breeding location and year effects.

Hope these comments help the authors improve the clarity of the geolocator analysis.
Aldina Franco

Reviewer #2 (Remarks to the Author):

Review of NCCC_290783

The manuscript on changes in migration of bar-tailed godwits, demonstrating that observed increases in the dates of migration are well within the range of individual-level variability in migration timing, is impressive, well written, and makes an important contribution! I enjoyed reading the manuscript, it provides much food for thought, and the deep understanding of the ecology of the species, and the various stages of their migration, makes for a very rich discussion of the observed patterns.

My comments fall into two main categories. The first is questions regarding the interpretation of the models. The second is in regards to the way the satellite data for the breeding grounds was analyzed. My background is in remote sensing, so those latter comments are the most detailed, and I have substantial concerns about the satellite data portion of the paper. Having said that, the satellite data analyses are a minor portion of the study, and the shortcomings can be overcome, so my critical assessment of the remote sensing portion should not detract from my overall impression of this manuscript, which was quite positive.

MODEL INTERPRETATION

A major goal of the statistical analyses was to “explicitly separate phenotypic flexibility from between-individual shifts in the population over time” (line 131). With all due respect, I do not think that that was possible with the dataset at hand, and the manuscript says so quite succinctly: “A longer time series is required to specifically address generational changes in this species” (line 149). In parts of the manuscript, the interpretation of the fact that there was no statistically difference among within- and between-individual slopes is rightfully conservation. For example, I much concur with the statement that “no selection or generation shift is required to explain our results” (line 132). However, other statements go too far, and I do not think it is valid to say that “This population-level change occurred through within-individual advancement” (line 22). Ultimately, the time-series that was analyzed was short, and variability among individual and within individuals was high. So yes, “within-individual advancement alone was sufficient to explain the trend”, but that does not mean that it is certain that individual-level advancement alone is what caused it, and I recommend going through the manuscript to make sure the wording does not give an impression that findings are more certain than they can be.

Related to this is the question of what constitutes a ‘long-term trend’. To some extent, that is a

philosophical question, and in the eye of the beholder, and yes, thirteen years of consistent data for a species and study system like this is impressive. However, thirteen years is also only half of the life span of bar-tailed godwits (line 152), and there are many multi-year climate cycles that could be at play. For example, the study period includes only two of three El Nino events. To be clear, I'm not saying that El Nino necessarily matter, just saying that thirteen years are rather short to establish a long-term trend, especially in a species that is long-lived, and that breeds in the Arctic with its high inter-annual variability. My recommendation is to remove the word 'long-term', and to go through the manuscript to make sure that the wording does not give the impression that the observed changes are not necessarily part of a trend that goes beyond the period that was observed.

SATELLITE DATA ANALYSES

The analysis of snowmelt is based on an excellent dataset, but I was unsure about the model fitting. In the example given in Fig. S4a, the switch from snow cover to snow melt occurs only once in spring, and once in the fall. While there may be individual pixels for which that is the case, I expect that there are many others that switch back and forth a few times during the shoulder seasons. The use of a threshold of "1/3 of the pixel snow covered" suggests that the fitted curve was not always as steep as in the example pixel. It would be nice to see an example of the model fit for such a pixel, and more detailed information how often there were multiple changes from snow cover to snow melt and back during spring, and how that may have affect the trend estimates.

The analyses of NDVI baffled me. I'm not saying that they are wrong, but choices of both NDVI dataset and algorithms to calculate phenology were surprising, to say the least, and lacked justification. In terms of the dataset, it was not clear to my why AVHRR data was analyzed instead of MODIS data. Both provide NDVI data for the entire study period, but the MODIS data has higher spatial resolution and is better calibrated. At the very least, there ought to be some justification why the inferior AVHRR data was chosen.

In terms of algorithms to calculate phenology, the combination of a cosine function (to separate winter and summer), followed by the fitting of the asymmetric double-sigmoid curve, is something I had not encountered before, and I was missing any references for this approach. There is a large remote sensing literature on how to estimate phenological data, such as the start of the growing season, but that literature was not referenced and it was not clear to my how the approach presented here differs, and if it is better. Again, at the least, I would like to see some justification why this approach was chosen.

I was also rather confused by the splitting of the year into 'winter' and 'summer', and then fitting curves separately. First, northern Alaska is affected by polar night, and so there are no observations for parts winter, and it was not clear what was modeled. Second, even in southern Alaska most observations are likely snow in winter. Setting them to zero makes sense, but again, there can not be many actual observations to model for winter. Third, the start of the growing season is bound to occur right at the end of winter or the beginning of summer. When fitting any curve, the fit is poorest at the edge of the range, so by splitting the curves the estimate of the start of the growing season would be worse. Last but not least, when looking at Fig S4b, it appears that one model fit is for the entire year. As I said, I was confused, and missing at the very least justification and better explanation.

Having said that regarding the phenology data source and how it was processed, my strong recommendation would be to download MODIS MCD12Q2 Land Cover Dynamics data and redo the

analyses based on the date of the start of the growing season that is readily available in that product. This would not be difficult to do, and while results may turn out to be fairly similar, I think that readers with remote sensing knowledge would trust MODIS based results much more because the currently chosen dataset and algorithms are inferior and unproven, respectively.

Reviewer #3 (Remarks to the Author):

By triangulating data from a range of high quality field studies and analysis of remote sensing data, the authors show convincingly in this paper that individual plasticity in migration timing is sufficient to explain long term trends in departure dates of bar-tailed godwits from New Zealand. The authors outline several possible mechanisms that might be driving these changes in migration timing, and elegantly parse out how the change in departure date from New Zealand is reflected in arrival on the stopover and breeding grounds. Unexpectedly, the results show that birds arrive earlier at the stopover sites in East Asia, and spend longer there, ultimately arriving no earlier on the breeding grounds as the time series progresses.

This study immaculately weaves together multiple streams of evidence in a way that is compelling, yet doesn't oversell the results or stretch the conclusions beyond the data. The writing is beautiful. Studying animal migration is intensely difficult, especially the connections between individual plasticity and population change, and I regard this paper as a major step forward both in terms of behavioural / ecological understanding of migration timing, but also how individuals (and ultimately populations) might respond to anthropogenic impacts. As such, this paper will be of interest to a very broad range of readers across the natural sciences from physiologists and migration specialists to those interested in the conservation of migratory species and global change biology more broadly.

I have only two comments on the work.

First, much of the underpinning logic early in the manuscript is built on the idea of climate-change-drives-earlier-migration. I wonder if the introduction might be more generally applicable if it could make clear that changes in the timing of various migration components could occur in response to a range of global change phenomena, including, but not limited to, climate change.

Second, line 203-205: Is copying of migration timing possible? I'd like to hear more of the authors' thoughts about this. The text mentions copying on the "first migration", but do juveniles on their first southward migration have the opportunity to copy adults? Or is the text referring to young birds copying the timing of the first northward migration from the adults around it? Perhaps this could be made a little clearer, since some sort of copying mechanism seems important for individual plasticity to drive these changes.

One final, minor comment: "Population advancement" is a rather obscure term for the title, and I don't think it would be immediately clear to many readers what that meant. The concept is rather quickly and clearly defined as the manuscript gets underway, but I wonder if there are alternative formulations of words that could be used in the title?

Richard Fuller, University of Queensland

Author responses in bold. Line numbers [xx–xx] refer to the tracked-change draft with ‘All Markup’ turned on.

REVIEWER COMMENTS

Reviewer #1 (Remarks to the Author):

This study is one of the first to robustly show that individual birds are changing their phenology of migration and departing from wintering areas earlier in recent years, coinciding with the patterns expected by climate warming and the need to arrive at breeding sites earlier. Bar-tailed godwits are long lived species and differences in phenology could result from individual or population responses to environmental change (previous studies have shown disparate results). This is a long term study quantifying both population and individual phenology changes using colour rings observations and geolocator information. The authors show that departure dates from wintering sites in New Zealand advanced 0.46d/y at the population level and this was consistent with the individual responses (0.42d/y). This study shows that in long lived species individuals have the capacity to adjust their migration phenology throughout their lives, this is an interesting result with implications for conservation as adaptation to environmental change can occur at the individual level rather than only through natural selection at the population level.

Reply: Thank you for the kind words.

A few questions remain (see below).

The population and individual data on departure from NZ was obtained from a long term dataset (13 years of data) and the results are robust. These results clearly show that phenology changes are occurring at the population and individual level, although there is high variability within individual trends “Within-individual trends ranged from –6.5 to +5.5 d/yr, and were advancing in 69% of individuals”, suggesting that there could be differences between northern and southern Alaskan breeders. The authors attempted to explain this variability based on geolocator information that provide full migration phenology information for individual godwits.

The geolocator information complements the individual analysis obtained for the departure from NZ dates and provides further insights into the timing of different stages of the migration. This dataset was obtained over 4 years (2 early and 2 later years) spread apart, hence is limited, and stochastic environmental variability could influence the results obtained.

Reply: We realize the limitation of the tracking data with respect to environmental variability. We have added a statement to this effect in the Discussion [239–240].

Furthermore, the proportion of south/north Alaskan breeding birds changes from the first to the second period of tracking which could influence the results. Lack of advancement in arrival date and start of breeding could be a reflection of a higher proportion of Northern breeders in the second period of tracking. To test if this is the case, the authors could have included the interaction between year and breeding location in the models presented in table

S3 (this may be stretching the model due to the small sample size, if this is the case a bootstrapping approach could be used).

Reply: We appreciate the reviewer raising this point. We agree that a changing proportion of northern vs. southern breeders could potentially confound the detection of a timing trend across years, and we had made sure this was not affecting our conclusions. Although we neglected to include this in the paper, we initially included the interaction of year*breeding region in our models. However, the interaction was non-significant for all six phenology parameters (and was actually weakest in the two Alaska timing parameters, which are most important for our conclusions), and so we excluded it from the final models reported in Table S3. We stand by this decision. However, to make this clear to readers, we have added a statement to this effect in the Methods [462–465].

Although the sample is indeed modest, we feel the power is sufficient to test for advancement in arrival in Alaska (i.e. to trust that a negative statistical result reflects a lack of advancement), as the same sample produced greater effect sizes and significant *p*-values for the first flight from New Zealand to the Yellow Sea. So, we consider the lack of advancement in arrival and breeding to be a robust finding.

In the discussion the authors give two hypotheses for the increasing stopover duration observed at the Yellow sea site. These are plausible but it would be good to know more about the length of the stay in New Zealand, is this also shifting? This may help understand the changes in phenology and raise alternative hypothesis for the early departure from NZ and longer stopover.

Reply: Arrival in New Zealand peaks in mid-late September, but is less individually consistent and much more protracted at the population level. Therefore, we cannot detect subtle shifts with the same precision as departure, but we have no evidence that the length of stay in New Zealand has changed during our study, and we have now added a statement to this effect [201–202]. However, we have previously shown that godwits do not appear particularly constrained by this six-month period, which is characterized by low predation pressure, slow fuelling amid low competition, and a lack of within- or between-season consequences for individual consistency of departure (Conklin & Battley 2012, Conklin et al. 2017), which we had briefly mentioned in the paper [199–200]. We don't think further speculation is warranted.

Another hypothesis: Godwits are social birds, if southern breeders are advancing their breeding date and migrating early, this could induce northern breeders to abandon wintering areas early (staying behind in smaller groups could lead to increased predation risk), but lead them to stay longer at the stopover locations compensating for the early departure. Stopover locations with larger numbers of birds from different wintering and breeding locations may provide good conditions to wait for the “right departure time”. If only southern breeders are advancing their migratory phenology, this would not be noticeable after the departure from NZ (again this could be tested by including the interaction between year and breeding location in the models presented in table S3).

Reply: If I understand the reviewer, the idea is that one group might be adaptively advancing their migration based on, for example, conditions in Alaska; meanwhile, the other group need not be similarly motivated, but might act similarly, at least through part of the migration, based solely on the behavior of the other group. It is

certainly good to recognize that social cues *per se* might additionally influence migration timing, but we have no data to specifically address this; however, we have added some words to the Discussion to better recognize this potential factor [215–217].

However, we have no evidence that this is driving differences among groups in our study, and feel that further speculation is unwarranted in this paper. One of the strengths of including geolocator-tracking data was to clarify what happened after New Zealand departure. In this case, neither group has advanced their departure from Asia or start of breeding, and, as discussed above, there were no significant interactions between year and breeding region. Therefore, it is most parsimonious to conclude that the observed changes are driven by what the groups have in common (i.e. staging area in the Yellow Sea) rather than what they do not (breeding phenology in Alaska).

Overall, I enjoyed reading this manuscript, it is clearly structured and provides new insights on how changes in migratory phenology operate in long lived species, at the population and individual level. The results presented are robust but the geolocator information could have been examined for interactions between breeding location and year effects.

Reply: We appreciate these comments, and trust we have sufficiently dealt with the issue of potential interactions.

Hope these comments help the authors improve the clarity of the geolocator analysis.
Aldina Franco

Reviewer #2 (Remarks to the Author):

Review of NCCC_290783

The manuscript on changes in migration of bar-tailed godwits, demonstrating that observed increases in the dates of migration are well within the range of individual-level variability in migration timing, is impressive, well written, and makes an important contribution! I enjoyed reading the manuscript, it provides much food for thought, and the deep understanding of the ecology of the species, and the various stages of their migration, makes for a very rich discussion of the observed patterns.

Reply: Much appreciated!

My comments fall into two main categories. The first is questions regarding the interpretation of the models. The second is in regards to the way the satellite data for the breeding grounds was analyzed. My background is in remote sensing, so those latter comments are the most detailed, and I have substantial concerns about the satellite data portion of the paper. Having said that, the satellite data analyses are a minor portion of the study, and the shortcomings can be overcome, so my critical assessment of the remote sensing portion should not detract from my overall impression of this manuscript, which was quite positive.

MODEL INTERPRETATION

A major goal of the statistical analyses was to “explicitly separate phenotypic flexibility from between-individual shifts in the population over time” (line 131). With all due respect, I do not think that that was possible with the dataset at hand, and the manuscript says so quite succinctly: “A longer time series is required to specifically address generational changes in this species” (line 149).

Reply: The reviewer is correct, in that our study is not able to directly test for generational change, and we made this point ourselves. But one of the strengths of our study system, and the most unique contribution of our study, is the characterization of within-individual changes against the entire range of population variation across 13 years, for which few studies have relevant data, and no studies have analyzed in a similar way. The within-subject centering analysis is specifically designed to separate and compare the contributions of within-individual variation (phenotypic flexibility) and between-individual changes (i.e. through replacement) during the same time period. The analysis demonstrated that within-individual changes occurred, and were in fact sufficient to explain the entire population trend. Therefore, we do not think our aim was over-stated or impossible, and we contend that we achieved it.

This is separate from the issue of whether phenotypic flexibility is the only tool available to godwits, or whether developmental or evolutionary processes would be detected or required over a much longer timeframe. This is the point we tried to make, but we don't think it undermines our results or claims.

In parts of the manuscript, the interpretation of the fact that there was no statistically difference among within- and between-individual slopes is rightfully conservative. For example, I much concur with the statement that “no selection or generation shift is required to explain our results” (line 132). However, other statements go too far, and I do not think it is valid to say that “This population-level change occurred through within-individual

advancement” (line 22). Ultimately, the time-series that was analyzed was short, and variability among individual and within individuals was high. So yes, “within-individual advancement alone was sufficient to explain the trend”, but that does not mean that it is certain that individual-level advancement alone is what caused it, and I recommend going through the manuscript to make sure the wording does not give an impression that findings are more certain than they can be.

Reply: This is a fair concern, and we have found three such cases, and have reworded them [22, 85, 269].

Related to this is the question of what constitutes a ‘long-term trend’. To some extent, that is a philosophical question, and in the eye of the beholder, and yes, thirteen years of consistent data for a species and study system like this is impressive. However, thirteen years is also only half of the life span of bar-tailed godwits (line 152), and there are many multi-year climate cycles that could be at play. For example, the study period includes only two or three El Nino events. To be clear, I’m not saying that El Nino necessarily matter, just saying that thirteen years are rather short to establish a long-term trend, especially in a species that is long-lived, and that breeds in the Arctic with its high inter-annual variability. My recommendation is to remove the word ‘long-term’, and to go through the manuscript to make sure that the wording does not give the impression that the observed changes are not necessarily part of a trend that goes beyond the period that was observed.

Reply: Although we mention that godwits can live >25 years (much longer than our study), the average godwit in New Zealand will only live ca. 10–12 years and migrate 9 times to Alaska (less than the duration of our study). Our previous formulation tends to exaggerate the issue, and so we have reworked this passage to be more accurate [158–162].

However, the reviewer’s point is still valid, with regard to the duration of our time series relative to the life-history of our species and relevant climate cycles.

Several of our uses of the term ‘long-term’ were general to the literature, and important to contrast our approach to the many migration studies that use only 2-3 years of data to characterize within-individual variation. None of these cases specifically refer to our study as ‘long-term’, and we have left these unchanged.

In the Abstract, the one case of ‘long-term’ has been removed in the revised version.

We also used ‘long-term’ to differentiate our full study period from the shorter period for which we have geolocator tracking data. For these three cases [234, 472–473, 761], we either removed ‘long-term’ or changed it to ‘13-year’ to avoid over-stating our study period.

SATELLITE DATA ANALYSES

The analysis of snowmelt is based on an excellent dataset, but I was unsure about the model fitting. In the example given in Fig. S4a, the switch from snow cover to snow melt occurs only once in spring, and once in the fall. While there may be individual pixels for which that is the case, I expect that there are many others that switch back and forth a few times during the shoulder seasons. The use of a threshold of “1/3 of the pixel snow covered” suggests that the fitted curve was not always as steep as in the example pixel. It would be

nice to see an example of the model fit for such a pixel, and more detailed information how often there were multiple changes from snow cover to snow melt and back during spring, and how that may have affect the trend estimates.

Reply: That is indeed a good point that needs consideration and clarification in the methods. We have now compared three methods to identify the annual snow-melt date: 1) first snow-free day per year, 2) the snow-free day followed by at least 60 days of snow-free conditions, 3) the original gaussian fit with a 1/3 threshold. The idea of the gaussian fit is actually a compromise of the first day (1) and the complete free day (2) since the modeled slope is not as steep if there are days that are snow-free in the early season, followed by snow events. In such a case, the modeled 1/3 day is between (1) and (2). In fact, the three different measures are very similar in the majority of cases (clear transition between snow and snow-free conditions) but differ in some years (e.g. 2009, 2016, 2018). However, the trends are robust (see figure and table below).

Therefore, we have left the snowmelt results unchanged, except for an inconsequential recalculation of the slope SEs [101–105, Table S2]. However, we now provide more details in the Methods [480–483] and in supplemental code that can also be viewed here: [https://github.com/slisovski/BTGodwit Alaska SnowNDVI](https://github.com/slisovski/BTGodwit_Alaska_SnowNDVI)

Trend 2008-2020

	Breeding area	Northern subset	Southern subset
Gaussian fit	-0.651 d/yr (95% CI, -0.663, -0.639)	-0.235 (-0.248, -0.222)	-1.926 (-1.951, -1.900)
First day	-0.665 d/yr (95% CI, -0.678, -0.652)	-0.227 (-0.241, -0.213)	-2.007 (-2.033, -1.981)
Free day	-0.560 d/yr (95% CI, -0.572, -0.548)	-0.168 (-0.181, -0.156)	-1.756 (-1.781, -1.730)

Trend 2008-2014

	Breeding area	Northern subset	Southern subset
Gaussian fit	-0.174 d/yr (95% CI, -0.200, -0.147)	0.394 (0.367, 0.421)	-1.913 (-1.976, -1.849)
First day	-0.223 d/yr (95% CI, -0.251, -0.194)	0.342 (0.312, 0.373)	-1.954 (-2.017, -1.89)
Free day	-0.216 d/yr (95% CI, -0.242, -0.190)	0.287 (0.261, 0.314)	-1.757 (-1.82, -1.693)

The analyses of NDVI baffled me. I'm not saying that they are wrong, but choices of both NDVI dataset and algorithms to calculate phenology were surprising, to say the least, and lacked justification. In terms of the dataset, it was not clear to me why AVHRR data was analyzed instead of MODIS data. Both provide NDVI data for the entire study period, but the

MODIS data has higher spatial resolution and is better calibrated. At the very least, there ought to be some justification why the inferior AVHRR data was chosen.

Reply: We agree that the AVHRR dataset is inferior to the MODIS datasets. However, the dataset we used is actually a re-processed Vegetation Health data set derived from VIIRS (2013–present) and AVHRR (1981–2012) GAC data (thus, our naming “NOAA STAR AVHRR VHP” was incorrect and misleading). We used this dataset because (unlike MODIS) it covers the entire study period (2008–2020) and provides consistent noise-reduced validated [2] NDVI data on the same spatial scale as the IMS snow cover dataset [1] we used.

More information on the product: It was processed by the newly developed operational VHP system. The new VHP system was improved from GVI-x VH system and some changes/improvements were made to meet the requirement of operation and improve data quality. It can process GAC data from NOAA-19, as well as FRAC data from METOP-A and METOP-B. It also produces vegetation health products from VIIRS on NPP and JPSS satellites. The VHP system is operationally running at NOAA Office of Satellite and Product Operations (OSPO) and providing official VHP products. This website provides recent VH data as a backup/alternative data source. VHP product posted on this VH web site should be consistent to that released by OSPO.

[1] https://www.star.nesdis.noaa.gov/smcd/emb/vci/VH/vh_ftp.php

[2] https://www.star.nesdis.noaa.gov/smcd/emb/vci/VH/vh_validation_AUSwheat.php

In terms of algorithms to calculate phenology, the combination of a cosine function (to separate winter and summer), followed by the fitting of the asymmetric double-sigmoid curve, is something I had not encountered before, and I was missing any references for this approach. There is a large remote sensing literature on how to estimate phenological data, such as the start of the growing season, but that literature was not referenced and it was not clear to me how the approach presented here differs, and if it is better. Again, at the least, I would like to see some justification why this approach was chosen.

I was also rather confused by the splitting of the year into ‘winter’ and ‘summer’, and then fitting curves separately. First, northern Alaska is affected by polar night, and so there are no observations for parts winter, and it was not clear what was modeled. Second, even in southern Alaska most observations are likely snow in winter. Setting them to zero makes sense, but again, there can not be many actual observations to model for winter. Third, the start of the growing season is bound to occur right at the end of winter or the beginning of summer. When fitting any curve, the fit is poorest at the edge of the range, so by splitting the curves the estimate of the start of the growing season would be worse. Last but not least, when looking at Fig S4b, it appears that one model fit is for the entire year. As I said, I was confused, and missing at the very least justification and better explanation.

Reply: We indeed failed to provide references for the method. This method was described and evaluated in Bradley et al. 2007 [1] and, based on our own experience, the method provides reliable results for arctic environments. Nevertheless, it seems overcomplicated and maybe outdated. The estimated winter values (zeros) are also problematic to some extent.

We have therefore simplified the methods and used more flexible smoothing techniques to model the NDVI dynamics (based on Bolton et al. 2020 [2]). In a nutshell, we used the above-described NOAA STAR dataset providing weekly noise-reduced NDVI values on a 4x4 km resolution. To identify the start of the season (15% threshold) we created a time series at each pixel and each year at a daily time step using penalized cubic smoothing splines. For a robust fit, we used the NDVI values for all surrounding cells within a radius of 15 km. The smoothing spline was, however, fitted by weighting the NDVI values according to the distance to the focal cell (Gaussian decline modeled as the positive side of a normal density distribution with a standard deviation of 4 km). Daily values of the smoothed spline were only predicted for the “summer” period when the area (all pixels within the 24 km radius) was not completely covered with snow.

We revised the Methods [493–506] to reflect this, and provide detailed description and code in the supplement as well as online:

https://github.com/slisovski/BTGodwit_Alaska_SnowNDVI

Also, these changes are reflected in the Results [106–107], Fig. 3d, and Table S2.

[1] Bradley BA, Jacob RW, Hermance JF, & Mustard JF (2007) A curve fitting procedure to derive inter-annual phenologies from time series of noisy satellite NDVI data. *Remote Sens Environ* 106(2):137-145.

[2] Douglas K. Bolton, Josh M. Gray, Eli K. Melaas, Minkyu Moon, Lars Eklundh, Mark A. Friedl (2020) Continental-scale land surface phenology from harmonized Landsat 8 and Sentinel-2 imagery. *Remote Sens Environ*, Vol. 240, <https://doi.org/10.1016/j.rse.2020.111685>.

Having said that regarding the phenology data source and how it was processed, my strong recommendation would be to download MODIS MCD12Q2 Land Cover Dynamics data and redo the analyses based on the date of the start of the growing season that is readily available in that product. This would not be difficult to do, and while results may turn out to be fairly similar, I think that readers with remote sensing knowledge would trust MODIS based results much more because the currently chosen dataset and algorithms are inferior and unproven, respectively.

Reply: Unfortunately, it seems that the MODIS MCD12Q2 Land Cover Dynamic Dataset does not cover the entire study period, since it does not provide data for 2019 and 2020. However, we have explored the dataset and compared the green-up dates with our green-up estimates from the NOAA STAR NDVI product. While the median values for the years are very similar (see figure below), the mean values differ quite significantly, with lower annual values for the MODIS MCD12Q2 dataset. In fact, many pixels have green-up dates that are well before the snow melt date (February, March, April). This can obviously be due to the different resolutions and that small (500 m) patches are snow-free during the early season. Most importantly, the trends and especially the differences in trends between southern and northern regions are robust between the datasets. We have included a short summary of the comparison and hope that the remote sensing community appreciates the explanation and can trust the general trends given that they are consistent among different dataset and methods.

Detailed description and methods and code can be found here:

https://github.com/slisovski/BTGodwit_Alaska_SnowNDVI

Comparisons between the MODIS greenup dates (500 m spatial resolution) and the greenup dates we estimated based on the NOAA STAR NDVI product (4 km spatial resolution).

	Breeding area	Northern subset	Southern subset
MOD	-1.042 d/yr (95% CI, -1.203, -0.881)	-0.684 d/yr (-0.845, -0.524)	-2.927 d/yr (-3.078, -2.777)
NOAA	-0.702 d/yr (95% CI, -0.716, -0.687)	-0.051 d/yr (-0.065, -0.036)	-2.533 d/yr (-2.563, -2.503)

Reviewer #3 (Remarks to the Author):

By triangulating data from a range of high quality field studies and analysis of remote sensing data, the authors show convincingly in this paper that individual plasticity in migration timing is sufficient to explain long term trends in departure dates of bar-tailed godwits from New Zealand. The authors outline several possible mechanisms that might be driving these changes in migration timing, and elegantly parse out how the change in departure date from New Zealand is reflected in arrival on the stopover and breeding grounds. Unexpectedly, the results show that birds arrive earlier at the stopover sites in East Asia, and spend longer there, ultimately arriving no earlier on the breeding grounds as the time series progresses.

This study immaculately weaves together multiple streams of evidence in a way that is compelling, yet doesn't oversell the results or stretch the conclusions beyond the data. The writing is beautiful. Studying animal migration is intensely difficult, especially the connections between individual plasticity and population change, and I regard this paper as a major step forward both in terms of behavioural / ecological understanding of migration timing, but also how individuals (and ultimately populations) might respond to anthropogenic impacts. As such, this paper will be of interest to a very broad range of readers across the natural sciences from physiologists and migration specialists to those interested in the conservation of migratory species and global change biology more broadly.

Reply: Thank you very much!

I have only two comments on the work.

First, much of the underpinning logic early in the manuscript is built on the idea of climate-change-drives-earlier-migration. I wonder if the introduction might be more generally applicable if it could make clear that changes in the timing of various migration components could occur in response to a range of global change phenomena, including, but not limited to, climate change.

Reply: We agree that the topic of 'plasticity vs evolution' in adaptive change is applicable far beyond just climate change, and in fact our results imply non-climate-related impacts. We had previously opted to simplify the Introduction for the sake of brevity, but we see the strength in bringing these points forward from the Discussion. We have reworked the first paragraph of the Introduction to place climate-related phenology in a more general context [32–39].

Second, line 203-205: Is copying of migration timing possible? I'd like to hear more of the authors' thoughts about this. The text mentions copying on the "first migration", but do juveniles on their first southward migration have the opportunity to copy adults? Or is the text referring to young birds copying the timing of the first northward migration from the adults around it? Perhaps this could be made a little clearer, since some sort of copying mechanism seems important for individual plasticity to drive these changes.

Reply: Yes, we were referring to the first northward migration, when first-time young birds join flocks that are mostly composed of experienced adults. We have added more detail to this paragraph to clarify our thoughts on this [215–219].

One final, minor comment: "Population advancement" is a rather obscure term for the title,

and I don't think it would be immediately clear to many readers what that meant. The concept is rather quickly and clearly defined as the manuscript gets underway, but I wonder if there are alternative formulations of words that could be used in the title?

Reply: We were trying to include the concept of comparing population and individual data in the title, as that is a unique contribution of our paper, but we agree that it doesn't necessarily communicate to non-specialists. We have simply removed 'Population' from the title, and also clarified the term 'advancement' in the first line of the Abstract by replacing 'timing of bird migration is advancing' to the more digestible 'bird migration is occurring earlier' [17].

Richard Fuller, University of Queensland

Reviewer comments, second round –

Reviewer #1 (Remarks to the Author):

No more comments, I am happy with the revised version.

Reviewer #2 (Remarks to the Author):

The authors have conducted substantial new analyses to address my concerns regarding the remote sensing analyses, and they have modified the text according to all of my other suggestions nicely. I appreciate that the way the authors responded and reacted to my comments, and have no further suggestions. This is an exciting study, and I commend the authors on a very nice paper.
Sincerely, Volker Radeloff

REVIEWERS' COMMENTS

Reviewer #1 (Remarks to the Author):

No more comments, I am happy with the revised version.

Reviewer #2 (Remarks to the Author):

The authors have conducted substantial new analyses to address my concerns regarding the remote sensing analyses, and they have modified the text according to all of my other suggestions nicely. I appreciate that the way the authors responded and reacted to my comments, and have no further suggestions. This is an exciting study, and I commend the authors on a very nice paper.

Sincerely, Volker Radeloff

Response:

We thank both reviewers for their helpful and constructive input. Much appreciated!